# Genomic Landscape Alterations in Primary Tumor and Matched Lymph Node Metastasis in Hormone-Naïve Prostate Cancer Patients

**DOI:** 10.3390/cancers14174212

**Published:** 2022-08-30

**Authors:** Giorgio Ivan Russo, Paolo Bonacci, Dalida Bivona, Grete Francesca Privitera, Giuseppe Broggi, Rosario Caltabiano, Jessica Vella, Arturo Lo Giudice, Maria Giovanna Asmundo, Sebastiano Cimino, Giuseppe Morgia, Stefania Stefani, Nicolò Musso

**Affiliations:** 1Urology Section, Department of Surgery, University of Catania, 95125 Catania, Italy; 2Department of Biomedical and Biotechnological Sciences (BIOMETEC), University of Catania, 95125 Catania, Italy; 3Department of Medical and Surgical Sciences and Advanced Technologies “G. F. Ingrassia”, Anatomic Pathology, University of Catania, 95125 Catania, Italy; 4Department of Experimental Oncology, Mediterranean Institute of Oncology (IOM), 95125 Catania, Italy

**Keywords:** next generation sequencing, prognosis, outcome, radical prostatectomy, KIT, ABL1, HRAS, CTNNB1, ERBB4

## Abstract

**Simple Summary:**

The aim of the present study is to investigate genetic changes associated with lymph node metastasis in a cohort of hormone-naïve prostate cancer patients. We found mostly concordance between both primary PCa samples and matched lymph node metastasis in terms of genomic alteration. Indeed, the analysis of hotspot regions in genes such as *ERBB4*, *AKT1*, *FGFR2* and *MLH1* underline that specific alterations in the primary tumor are extremely important for cancer prognosis prediction.

**Abstract:**

Background: Prostate cancer (PCa) is a disease with a wide range of clinical manifestations. Up to the present date, the genetic understanding of patients with favorable or unfavorable prognosis is gaining interest for giving the appropriate tailored treatment. We aimed to investigate genetic changes associated with lymph node metastasis in a cohort of hormone-naïve Pca patients. Methods: We retrospectively analyzed data from 470 patients who underwent surgery for PCa between 2010 and 2020 at the Department of Urology, University of Catania. Inclusion criteria were patients with lymph node metastasis and patients with PCa with extra capsular extension (pT3) and negative lymph node metastasis. The final cohort consisted of 17 different patients (11 PCa with lymph node metastasis and 6 PCa without lymph node metastasis). Through the cBioPortal online tool, we analyzed gene alterations and their correlations with clinical factors. Results: A total of 688 intronic, synonym and nonsynonym mutations were sequenced. The gene with the most sequenced mutations was ERBB4 (83 mutations, 12% of 688 total), while the ones with the lower percentage of mutations were AKT1, FGFR2 and MLH1 (1 mutation alone, 0.14%). Conclusion: In the present study we found mostly concordance concerning the *ERBB4* mutation between both primary PCa samples and matched lymph node metastasis, underlining that the identification of alterations in the primary tumor is extremely important for cancer prognosis prediction.

## 1. Introduction

Prostate cancer (PCa) is a disease with a wide range of clinical manifestations. Every year, more than 900,000 new cases of PCa are diagnosed worldwide [1]. PCa can be kept under control in some cases, but in others it can progress into an aggressive form, generating metastasis and even leading to the patient’s death; in fact, PCa is still the second leading cause of cancer death all over the world [1]. 

At diagnosis time, the majority of PCa cases are either localized within the prostate gland or in the locoregional area, with invasion into surrounding structures or lymph nodes [2]. Metastasis at this stage has been associated with adverse prognosis and worse outcome [3]. However, some clinical data have reported that about 30% of patients with lymph node metastasis are recurrent-free after 10 years of follow-up, suggesting the importance of a better understanding of its heterogeneity [4]. 

In fact, a large percentage of PCa diagnoses may have a strong genetic component. In this regard, PCa risk has been linked to several single-gene mutations. These include BRCA1 and BRCA2 (breast cancer 1 and 2), ATM (ataxia telangaictesia-mutated) or HOXB13 (Homebox B13) [5,6]. Interestingly, Kneppers et al. compared the copy-number alterations of primary PCa and pelvic lymph node metastases showing that in 23.3% the index primary tumor was not clonally related to the locoregional lymph node metastases [7], further underlining that the identification of alterations in primary tumor or metastasis maybe of extremely importance for cancer prognosis prediction. 

It is also important to consider that in the last years, many studies have highlighted the hypothesis that diabetes and hyperglycemia could be relevant for PCa development and progression and, in some cases, with worse recurrence over time [8]. In a previous study from Broggi et al., it has been reported that the androgen receptor (AR) expression was associated with increased risk of pathological aggressiveness (odds ratio [OR]: 2.2; *p* < 0.05) and insulin-receptor-α(IR- α) expression (OR: 5.7; *p* < 0.05) and that insulin growth factor-1 (IGF-1) expression was predictive of pathological aggressiveness (OR: 16.5; *p* = 0.017) in patients with PCa and diabetes. 

Up to the present date, the genetic understanding of patients with more advanced disease and local metastasis and to evaluate potential driver of PCa progression is required. 

Based on these premises, the aim of the present study is to investigate genetic changes associated with lymph node metastasis in a cohort of hormone-naïve PCa patients. 

## 2. Materials and Methods

We retrospectively analyzed data from 470 European-Caucasian patients who underwent surgery for PCa and lymph node dissection between 2010 and 2020 at the Department of Urology, University of Catania. Inclusion criteria were patients with lymph node metastasis and patients with PCa with extra capsular extension (pT3) and negative lymph node metastasis. The final cohort consisted of 17 different patients (11 PCa with lymph node metastasis and 6 PCa without lymph node metastasis). 

For each patient, a specialist genito-urinary pathologist marked the following hematoxylin and eosin-stained sections: primary PCa tissue with the highest grade, lymph node metastasis tissue (in positive patients) and a matched distant area with no cancer. 

Furthermore, paraffin-embedded blocks of primary tumor samples were used for the assembly of the tissue microarray using the Galileo tissue micro array CK3500 (Integrated System Engineering, Milan, Italy), as previously published [8,9]. Immunohistochemical analyses with anti-androgen receptor (AR) (ab74272; rabbit polyclonal, 1:1200 dilution) [8], anti-insulin receptor-α (IR-α) (ab5500; rabbit polyclonal, 1:1000 dilution) [8], anti-IR-β (ab69508; mouse monoclonal clone: C18C4, 1:1000 dilution) [8], anti-insulin growth factor receptor (IGF1-R) (ab39398; rabbit polyclonal, 1:50 dilution) [8] and anti-prostate specific membrane antigen (PSMA) (ab64082; rabbit monoclonal; clone: SP29, 1:100 dilution) [8] antibodies stainings (all from Abcam, Cambridge, UK) were performed as previously described using manufacturer instructions [8,9,10,11]. The scoring system included a combined analysis of staining intensity (IS) and percentage of immunoreactive cells (extent score; ES), as previously described [10,11]. 

### 2.1. Deparaffinization Procedure and DNA Extraction

The deparaffinization and the DNA extraction were carried out following the manufacturer’s practice provided by QIAGEN (QIAamp^®^ DNA FFPE Advanced Kit, Ref. 56604, QIAGEN, 40724 Hilden, Germany). The results of deparaffinization and DNA quantification are shown in Appendix A.

In this regard, the use of paraffin-embedded blocks from primary or metastatic tissue has been previously validated in the PROfound study [12]. Thirteen samples reported insufficient coverage in the first sequencing to conduct a satisfactory analysis, so they were subjected to quality control. This was carried out using the Illumina^®^ FFPE QC Kit (Ref. 15013664, Illumina, Inc., San Diego, CA 92122, USA) following the manufacturer’s instructions. The results are reported in Appendix A. Even if some samples reported an insufficient ΔCTs (cycle threshold), they were equally sequenced a second time, reporting a fulfilling coverage. 

### 2.2. NGS Sequencing

All solutions containing the extracted DNA were diluted to a 5 ng/µL concentration and then 20 ng (corresponding to 4 µL) of each sample were used for sequencing by next-generation sequencing (NGS). This was carried out in the molecular biology laboratory of the University of Catania on an Illumina MiSeq platform according to the manufacturer’s instructions provided in the AmpliSeq^TM^ Cancer HotSpot Panel v2 for Illumina^®^ (Ref. 20019161, Illumina, Inc., 92122, San Diego, CA, USA); in the end, we were able to sequence 2800 COSMIC mutations from 50 oncogenes and tumor suppressor genes. Indexes were provided with AmpliSeq^TM^ CD Indexes, Set A for Illumina^®^ (96 Indexes, 96 Samples) (Ref. 20019105, Illumina, Inc., 92122, San Diego, CA, USA). Denature and dilute libraries were obtained following the “Denature and Dilute Libraries Guide” protocol provided by Illumina^®^ (Document # 15039740 v10), choosing as loading concentration 9 pM. Finally, sequencing was performed using the MiSeq Reagent Kits v3 (Ref. 15043895, Illumina, Inc., 92122, San Diego, CA, USA). The creation of the sample sheet was accomplished by using the Local Run Manager v3 software and following the instructions in the Local Run Manager v3 Software Guide provided by Illumina. 

### 2.3. Bioinformatics Analysis

The data obtained from the sequencing were uploaded to the Illumina Basespace Sequence Hub platform as FASTQ format and analyzed with the “DNA Amplicon” application by Illumina, Inc. for the detection of mutations in the genes belonging to the panel. 

A FASTQ file is a text file that contains the sequence data from the clusters that pass filter on a flow cell (for more information on clusters passing filter, see the “Additional Information” section of this bulletin). If samples were multiplexed, the first step in FASTQ file generation is demultiplexing. Demultiplexing assigns clusters to a sample based on the cluster’s index sequence(s). After demultiplexing, the assembled sequences are written to FASTQ files per sample. If samples were not multiplexed, the demultiplexing step does not occur and, for each flow cell lane, all clusters are assigned to a single sample.

For a single-read run, one Read 1 (R1) FASTQ file is created for each sample per flow cell lane. For a paired-end run, one R1 and one Read 2 (R2) FASTQ file is created for each sample for each lane. FASTQ files are compressed and created with the extension *.fastq.gz (https://emea.support.illumina.com/content/dam/illumina-support/documents/documentation/software_documentation/local-run-manager/local-run-manager-generate-fastq-workflow-guide-100000003344-03.pdf (accessed on 28 August 2022) and https://www.illumina.com/content/dam/illumina-marketing/documents/informatics/basespace-informatics-suite-info-sheet.pdf (accessed on 28 August 2022)).

Further information regarding the mutations obtained, i.e., amino acid mutations, the impact on protein structure and dbSNP_IDs, was inserted into the online tool PROVEAN^®^ to provide predictions for any type of protein sequence variations including the following. Clinical considerations, population incidence (minor allele frequency [MAF]) and genomic information were obtained from https://www.ncbi.nlm.nih.gov/snp/ (accessed on 28 August 2022) and https://www.ncbi.nlm.nih.gov/clinvar/ (accessed on 28 August 2022). 

### 2.4. Multiparameter Genetic Score

For each patient, we calculated the multiparameter genetic score that was obtained considering the genetic landscape, the allelic frequency spread among population and the clinical features. For each nonsynonymous single nucleotide polymorphism database (dbSNP) sequenced and reported in the literature, we created three scores that take into account the tropism of the dbSNP, i.e., on which samples the mutation was detected (tropism score), the frequency at which the second most common allele occurs in a given population (MAF score), and its clinical significance (ClinVar score). We calculated the three scores for each of our patients according to the specific mutation found and sequenced to understand which factor contributed the most to the neoplasm growth. Each score scale ranges from a minimum of 10 (less severe condition) to a maximum of 150 (more severe condition). 

Once the three scores have been applied for each dbSNPs present in every single patient, three summations were calculated (one per single score: ∑tropism score, MAF score in patient and ∑clinical score). 

### 2.5. cBioPortal Analysis

To validate the finding of our study, we used cBioPortal (https://www.cbioportal.org/ (accessed on 14 July 2022)) to verify the impact of the found alterations on survival. cBioPortal provides a visual tool for the research and analysis of cancer gene data and helps cancer tissue and cytology research gain molecular data understanding of their genetics, epigenetics, gene expression and proteomics [13]. Through the cBioPortal online tool, we analyzed gene alterations and their correlations with survival. We downloaded datasets from cBioPortal for prostate cancer, which provides visualization, analysis and downloading of large-scale cancer genomics datasets [13,14].

### 2.6. Statistical Analysis 

Continuous variables are presented as median and interquartile ranges (IQR) and were compared by the Student independent *t*-test or the Mann–Whitney U test based on their usual or unusual distribution, respectively (normality of variables’ distribution was tested by the Kolmogorov–Smirnov test). Categorical variables were tested with the χ^2^ test. 

To establish a potential correlation among the three scores and the globally calculated score and other clinical (age, prostate specific antigen) and nonclinical (presence of lymph nodes metastasis (N), pathological stage and Gleason score), Pearson’s correlation analyses were performed for the 17 patients; in particular, the analysis took into account both the global score, deriving from each of the finding mutation, and the specific percentage score from each of the three summations. This correlation matrix is a statistical tool that measures the linear correlation between two variables: X and Y. The matric has a value range between +1 and −1, where +1 indicates total positive linear correlation, 0 no linear correlation and −1 total negative linear correlation. The correlation coefficient ranges from −1 to +1, where +1 implies the existence of a linear equation establishing a relationship between two factors (X and Y) that increase simultaneously. In addition to this, R-squared was also calculated, which measures the reliability of the linear relationship between the variables included in the model. Its value is between 0 (fully correlated variables) and 1 (unrelated variables).

For all statistical comparisons, a significance level of *p* < 0.05 was considered to show any substantial difference between groups. 

Data analysis was performed under the guidance of our statistics expert, using SPSS version 17 (Statistical Package for Social Science. SPSS Inc. Released 2008. SPSS Statistics for Windows, Version 17.0.) (SPSS Inc., Chicago, IL, USA). 

## 3. Results

In the whole study cohort, the median age (interquartile range) was 65.0 (62.0–68.0), the median PSA was 15.9 (9.37–40.0). The majority of our patients had an ISUP grade group ≥4 (70.58%) and a pathological stage ≥pT3 (94.12%). 

Table 1 lists baseline characteristics of patients. 

### 3.1. DNA Amplicon Sequencing in Healty, Primary Tumor and Metastasis Samples Showed Heterogeneity in the Mutations Found 

On average, 15 multiple nucleotide variants (MNV) were sequenced from each sample, for a total of 688 mutations in the panel’s gene hotspots and 39 nonsynonymous mutations associated with a dbSNP or COSMIC code analyzed. Furthermore, for five of these mutations any reference (dbSNP or COSMIC) was found in the literature. However, these five mutations were sequenced just once in one single patient. Moreover, these mutations were sequenced on a total of 25 genes. The distribution of the mutations analyzed is shown in Table 2, while the average quality parameters are reported in Appendix A.

The patient in whom most nonsynonymous dbSNPs were found was Patient 8 (15 dbSNPs) and this was also the patient with the higher number of somatic mutations, sequenced only in tumor or lymph node samples (8 dbSNPs in tumor and 2dbSNPs in lymph node sample). The patient with the minor number of dbSNPs was Patient 12, with only three dbSNPs and both germline, since they have been sequenced on healthy and tumor samples. The most diffused dbSNP sequenced was rs1042522 on *TP53* gene (16 patients out of 17 reported this specific mutation), while the less diffused are the rs1419499014 on the *ERBB2* gene, the rs990046031 on the *AKT1* gene, the rs587781474 on the *STK11* gene, the rs1800863 on the *RET* gene, the rs839541 on the *ERBB4* gene, the rs199906407 on the *FTL* gene, the rs780807237 on the *PIK3CA* gene, both the rs28934578 and the rs587782006 on the *TP53* gene, the rs397772062 on the *KDR* gene, the rs34323200 on the *NPM1* gene, both the COSM5990839 and rs121913329 on the *APC* gene, the rs149119664 on the *FGF3* gene, the rs876659657 on *MLH1* and finally both the rs752729752 and the rs75580865 on the *FLT3* gene. All of these mutations were identified only in one sample out of 44. To extract not only germline variants, but also the somatic ones, the fastq patients’ samples were compared in pair for each patient. In fact, tumoral versus healthy tissues and lymph node versus healthy samples have been analyzed and compared. 

### 3.2. Multiparameter Genetic Score (MGS) Calculation Showed a Correlation with Lymph Node Metastasis

All three scores that are part of MGS have been calculated. The average quality data for all samples is reported in Appendix A. Almost all the patients showed a coherent trend: the MAF score was the major component in all 17 patients (MAF score ≥ 41.5 always). The ClinVar score was the second major component (18.3 ≤ ClinVar score ≤ 38.7) in 12 patients out of 17, while the tropism score was the minor component in almost all patients (12 out of 17, 6.7 ≤ tropism score ≤ 33.7). The average values showed that the MAF score was the major constituent of all MGSs, while the ClinVar and tropism scores were, respectively, about 1/4 (26.5%) and 1/5 (20.4%). 

We demonstrated that MAF was negatively associated with the tropism score (r = −0.8; *p* < 0.01) and slightly with lymph node metastasis (r = −0.45; *p* = 0.06); MGS was slightly correlated with the tropism Score (r = 0.43; *p* = 0.08) and the presence of lymph node metastasis was negatively associated with the tropism Score (r = −0.69; *p* < 0.01) (Figure 1 and Table 3). 

### 3.3. Total Impact of dbSNPs on AmpliSeq Pane Showed That ERRB4 Was the Most Frequent Mutation

A total of 688 intronic, synonym and nonsynonym mutations were sequenced. These mutations were sequenced on 32 different genes (64% of the 50 genes included in the panel). The gene with the most sequenced mutations was *ERBB4* (83 mutations, 12% of 688 total), while the least mutated were *AKT1*, *FGFR2* and *MLH1* (1 mutation alone, 0.14% of 688 total). The gene that reported the most nonsynonymous mutations was the *TP53* gene, where a total of 42 mutations were sequenced and all were nonsynonymous. Appendix A assumes all mutations sequenced. 

### 3.4. KIT, HRAS and CTNNB1 Alterations in a Combination Is Associated with Worse Surival

We further analyzed using the cBioPortal online tool genomic alterations that were associated with worse survival. In this regard, we studied those alterations that were expressed both in the primary tumor and in matched lymph node metastasis in our cohort. Consequently, we analyzed *ERBB4*, *HRAS*, *KIT*, *ABL1* and *CTNNB1* mutations and their correlation with prognosis using 23 selected studies (9377 samples). 

*ERBB4* was muted in 1.7% of cases, *KIT* in 1.3%, *ABL1* in 1.4%, *HRAS* in 1.3% and *CTNNB1* in 4% (Appendix A). 

Using data from cbBioPortal, when incorporating *KIT*, *HRAS* and *CTNNB1* alterations in a combination score we found that overall survival was statistically worse (*p* < 0.01) and also disease-free survival (*p* = 0.02) in respect to the unaltered group (Figure 2). 

We did not find significant association regarding overall survival and disease-free survival in positive patients at *ERBB4* (Figure 3) and *ABL1* (Figure 4). On the contrary, we found significant association between *KIT* alteration, overall survival and disease-free survival (Figure 5), between *HRAS* and overall survival (Figure 6) and a trend between *CTNNB1* and overall survival (Figure 7). 

### 3.5. Tissue Micro Array Analysis and Immunohistochemistry Stainings Showed an Association between IGF1-R Expression and ERRB4 Alterations in Primary Tumor or Lymph Node Metastasis

To verify the potential impact of glycemic metabolism and genomic alterations, we performed immunohistochemistry analysis for AR, IR-α, IR-β, IGF1-R and PSMA expression in the primary tumor. 

Regression analysis demonstrated weak statistical association between IGF1-R expression and ERRB4 alterations (r = 0.005; *p* < 0.01). Univariate logistic regression analysis showed strong association between IGF1-R expression and ERRB4 alterations in the primary tumor or lymph node metastasis (odds ratio: 54.0; *p* < 0.01). For the AR, IR-α, IR-β and PSMA expression, we did not find statistical associations. Finally, we did not perform logistic analysis for *KIT, HRAS, ALB1* or *CTNNB1* because of the low number of events in our small cohort. Appendix A shows the expression of all staining in prostate cancer. 

## 4. Discussion

Herein, in the present study, we reported genomic alterations in primary prostate cancer tissue and matched lymph node metastasis. Interestingly, concerning the analysis of prostate cancer sample and matched lymph node metastasis, we mostly found concordance between the two samples, underlining that the identification of alterations in the primary tumor is extremely important for cancer prognosis prediction. Interestingly, we found mostly concordance concerning the *ERBB4* mutation between both primary PCa samples and matched lymph node metastasis and that its expression was highly associated with IGF1-R staining. Furthermore, by demonstrating that the presence of lymph node metastasis is associated with low MAF (minor allele frequency) and low tropism, we underlined that less common genomic alterations may favor a more aggressive PCa disease. This regards, in fact, a wide genomic analysis of different stages of PCa. 

Interestingly, the Cancer Genome Atlas (TCGA) showed that 13 genes were recurrently mutated in prostate cancer: deletions of *SPOP*, *TP53*, *FOXA1*, *PTEN, MED12* and *CDKN1B*; additional clinically relevant genes were identified with lower frequencies, including *BRAF*, *HRAS*, *AKT1*, *CTNNB1* and *ATM* [15]. These findings are also corroborated by our present study, which showed that the combination of *KIT*, *HRAS* and *CTNNB1* alterations was associated with worse overall survival (*p* < 0.01) and also disease-free survival (*p* = 0.02), in respect to the unaltered group (Figure 2).

Although we did not assess molecular pathways underling such alterations and the impact of IGF1-R on these genomic findings, Worthington et al. reported that IGF1-dependent signaling and proliferation were enhanced in ErbB2-overexpressing cells and with increased invasiveness and anchorage-independent colony formation in breast cancer [16]. 

An interesting study by Zheng et al. investigated the role of 37 genes to predict lymph node invasion. The results of the RNA sequence in this study showed that 18 of 37 genes exhibited dysregulated expression between PCa and lymph node invasion samples, indicating that dysregulated expression levels of different genes played an important role in the lymph node invasion of PCa [17]. Among these, an androgen receptor (AR) may play an important role for disease progression [18]. 

A similar well-designed study aimed at identifying the genes associated with the involvement of lymph node metastasis in patients with PCa and, among 376 genes investigated, three genes, *RALGPS1*, *ZBTB34* and *GOLGA1*, had a significant copy number alteration [19]. 

Pudova et al. performed a bioinformatic analysis of the Cancer Genome Atlas (TCGA) data and RNA-Seq profiling of a Russian patient cohort to reveal prognostic markers of locally advanced lymph node-negative prostate cancer. Authors found different genes that were associated with favorable and unfavorable prognoses based on the TCGA (*B4GALNT4*, *PTK6* and *CHAT*) and Russian patient cohort data (*AKR1C1* and *AKR1C3*). Furthermore, authors revealed such genes for the *TMPRSS2-ERG* prostate cancer molecular subtype (*B4GALNT4, ASRGL1, MYBPC1, RGS11, SLC6A14, GALNT13* and *ST6GALNAC1*) [20].

Similarly, Schmidt et al. analyzed laser micro-dissected primary PC foci (*n* = 23), adjacent normal prostate tissue samples (*n* = 23) and lymph node metastases (*n* = 9) from 10 hormone-naïve PC patients. Genes important for PC progression were identified using differential gene expression and clustering analysis and they reported a list of 20 transcripts (2 upregulated, 18 downregulated). The seeding model significantly predicted time to biochemical recurrence independently of clinicopathological variables in uni- and multivariate Cox regression analysis in TCGA (univariate: HR 2.39, *p* < 0.0001, multivariate: HR 2.01, *p* < 0.0001) [21]. 

Several studies have investigated the genomic characteristics of metastatic hormone-sensitive PCa (mHSPC). A recent systematic review by Van der Eecken et al. was conducted on 11 studies that included 1682 mHSPC patients. A comparative analysis of gene alteration frequencies across disease states revealed a relative increase from localized to castration-resistant tumors, with noteworthy enrichment of *CTNNB1* alterations in mHSPC (5%) [22]. 

In fact, patients with PCa with alterations in canonical *WNT* pathway genes, which lead to β-catenin activation, are refractory to androgen receptor-targeted therapies, underlying that this genomic alteration may harbor a more aggressive cancer [23]. 

These results were confirmed by Isaacsson Velho, who showed that the different types of Wnt- pathway mutations (inactivating *APC* or *RNF43* mutations vs. activating *CTNNB1* mutations) were independently associated with higher hazard of PSA progression than Wnt wild type (aHR 2.58, *p* = 0.023). Despite a strong trend in the same direction, *CTNNB1* mutations showed no statistically significant association with higher hazard ratio (HR) of PSA progression (HR 2.12, *p* = 0.072) [24].

Generally, cancer has been linked to mutations in the *ERBB4* gene [25], which in this study was confirmed as a highly unstable gene. No information was found on the sequenced P.Q264* missense mutation in this gene, even if this has been already sequenced in a colon cancer cell line [26]. Regarding *CSF1R*, as for the SNP rs386693509, no MAF is reported in the databank for this mutation, but it has been sequenced on 15 patients out of 17 in our cohort (88.2%). Even if present in the literature, this mutation is very little discussed and has an unknown pathogenic impact [27]. In the *FGFR* gene, rs17881656 is including in a retrospective study that tested the NGS of selected cancer-associated genes in resected prostate cancer [28], while rs149119664 is not reported in the literature, even if classified as pathogenic (score by 0.97). 

Despite the confirmation of these interesting results, it is important to underline that it is not easy and reliable to perform such analysis in standard patients during clinical settings, since we have also to consider the dynamics of tumors and the long follow-up to be fulfilled. In this context, liquid biopsies may overcome the limitations of intrapatient tumor heterogeneity and of tissue biopsies, allowing for monitoring PCa disease [29,30].

Although there are no technology available to detect circulating tumor cells in all their phenotype and dynamic processes, new platforms and further studies are, however, ongoing to overcome all limitations [31]. In this context, the analysis of common alterations that are matched with lymph node metastasis can be helpful for cancer prognosis and treatment. 

Before concluding, we would like to underline some limitations. First, the discovery data were small since we included 17 patients and 45 samples. Furthermore, we applied a standard cancer hotspot and we did not assess further alterations. It is important to underline that more studies should implement the relationship between genomic alterations and androgen receptors, which is considered an important driver for PCa progression. Moreover, the MGS score was not internally and externally validated in the literature. Finally, we were not able to perform a multivariate model in our cohort because of the small sample size and, consequently, the few events. 

## 5. Conclusions

In the present study, we reported genomic alterations in the primary tumor and matched lymph node metastasis from hormone-naïve prostate cancer patients. We revealed that the gene with the most sequenced mutations was *ERBB4* (83 mutations, 12% of 688 total), while the least mutated were *AKT1*, *FGFR2* and *MLH1* (1 mutation alone, 0.14% of 688 total). We also found mostly concordance concerning the *ERBB4* mutation between both primary PCa samples and matched lymph node metastasis, underlining that the identification of alterations in the primary tumor is extremely important for cancer prognosis prediction. Furthermore, the *ERBB4* mutation is associated with greater IGF1-R staining, underling a putative link with glycemic control and PCa progression in this category of patients. 

Analyzing data from cBioPortal, we demonstrated worse overall survival and disease-free survival in patients with combined alteration of *KIT*, *HRAS* and *CTNNB1*. These results can be applied for monitoring disease and drug response after curative treatment. 

## Figures and Tables

**Figure 1 cancers-14-04212-f001:**
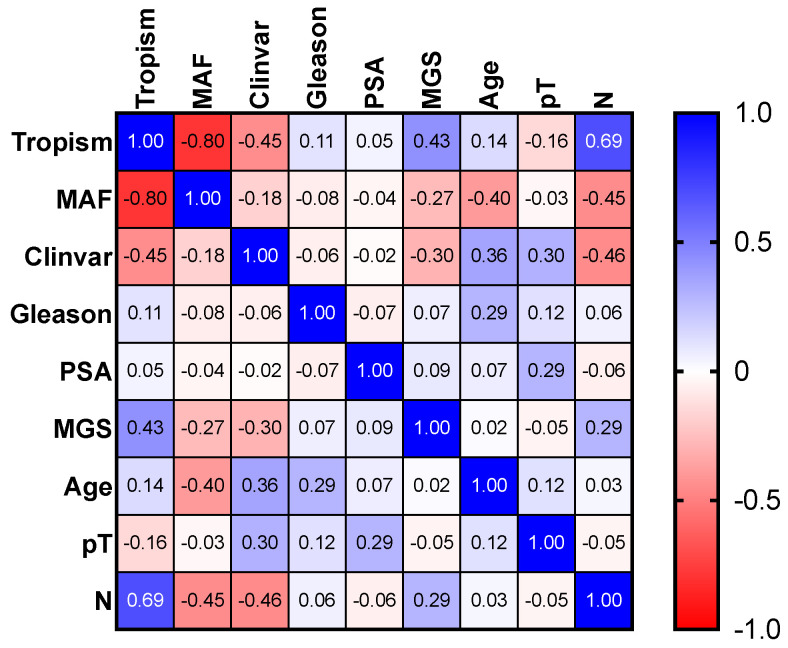
Correlation matrix (above) and *p*-values (under) reported for the correlation analysis. In the correlation matrix, the red color represents a complete negative correlation, while the blue color represents a complete positive correlation. The 0 value represent absence of correlation between the two variables. MAF = minor allele frequencies; Clinvar = clinical stage; MGS = multiparameter genetic score; pT = pathological stage; N = pathological lymph node metastasis; MAF = minor allele frequency.

**Figure 2 cancers-14-04212-f002:**
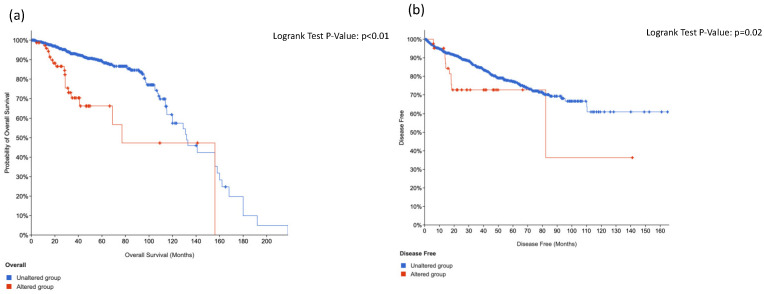
*KIT*, *HRAS* and *CTNNB1* combination score prognosis in prostate cancer for overall survival (**a**) and disease-free survival (**b**).

**Figure 3 cancers-14-04212-f003:**
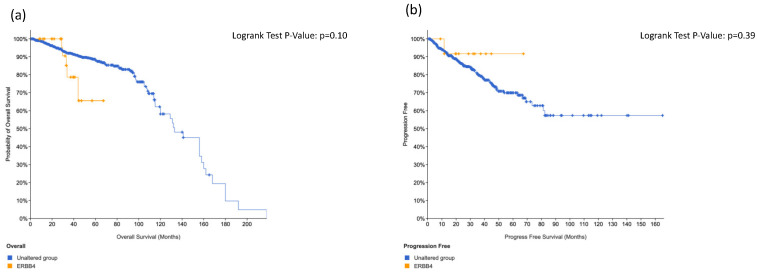
Overall survival (**a**) and disease-free survival (**b**) in ERRB4-positive prostate cancer patients.

**Figure 4 cancers-14-04212-f004:**
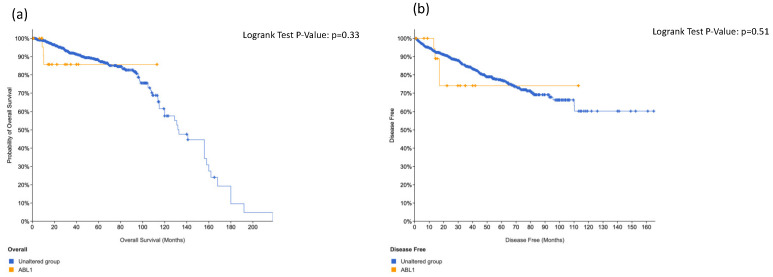
Overall survival (**a**) and disease-free survival (**b**) in ABL1-positive prostate cancer patients.

**Figure 5 cancers-14-04212-f005:**
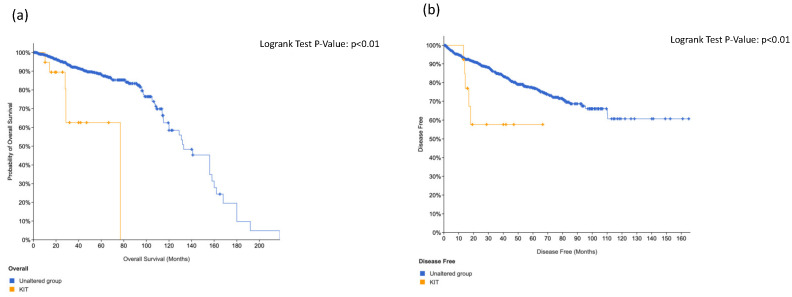
Overall survival (**a**) and disease-free survival (**b**) in KIT-positive prostate cancer patients.

**Figure 6 cancers-14-04212-f006:**
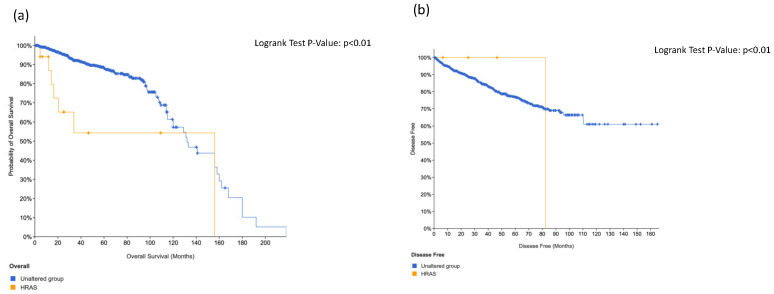
Overall survival (**a**) and disease-free survival (**b**) in HRAS-positive prostate cancer patients.

**Figure 7 cancers-14-04212-f007:**
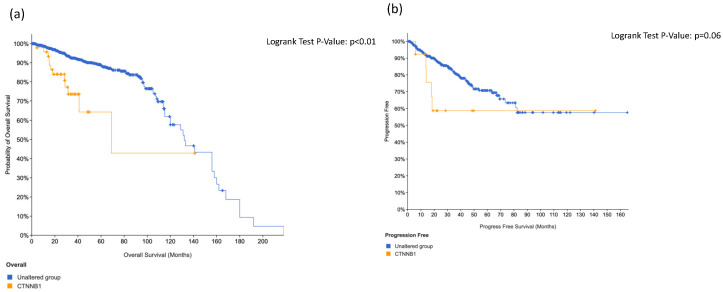
Overall survival (**a**) and disease-free survival (**b**) in CTNNB1-positive prostate cancer patients.

**Table 1 cancers-14-04212-t001:** Baseline characteristics of the patients.

Age (years), median (IQR)	65.0 (62.0–68.0)
PSA (ng/mL), median (IQR)	15.9 (9.37–40.0)
**Clinical stage, n (%)**	
T1–T2	14 (82.35)
T3	3 (17.65)
**Biopsy ISUP grade group, n (%)**	
1	1 (5.88)
2	6 (35.29)
3	2 (11.76)
4	6 (35.29)
5	2 (11.76)
**Pathological stage, n (%)**	
T2	1 (5.88)
T3a	5 (29.41)
T3b	11 (64.71)
**Pathological ISUP grade group, n (%)**	
2	1 (5.88)
3	4 (23.53)
4	2 (11.76)
5	10 (58.82)
**Pathological lymph node metastasis, n (%)**	
N0	6 (35.3)
N1	11 (64.7)

IQR = interquartile range; PSA = prostate specific antigen; ISUP = International Society of Urological Pathology.

**Table 2 cancers-14-04212-t002:** Summary of sequenced mutations associated with a dbSNP code. At the top is the number of the patient with the respective samples (H = healthy tissue, T = primary tumor, M = metastatic lymph node). On the left are the mutations and genes on which they are placed. On the right, minor allele frequencies (MAF) are also reported.

		Patients																																												
		1			2			3			4		5			6			7			8			9			10			11			12		13		14		15		16		17		MAF
**Gene**	**dbSNP**	H	T	M	H	T	M	H	T	M	H	T	H	T	M	H	T	M	H	T	M	H	T	M	H	T	M	H	T	M	H	T	M	H	T	H	T	H	T	H	T	H	T	H	T	
*ERBB2*	*rs1419499014*																																				x									0.000004
*ERBB4*	*rs67894136*	x	x	x	x	x	x					x		x	x			x		x	x	x	x	x		x	x		x	x	x	x	x						x	x	x	x	x		x	0.24
	*p.Q264* *								x																																					-
	*rs839541*														x																															0.3
*FGFR1*	*p.A268V*																																				x									-
*CSF1R*	*rs386693509*	x	x	x	x	x	x	x	x	x	x	x	x	x		x	x	x	x		x				x	x	x				x	x	x	x	x	x	x	x	x			x	x	x	x	-
*TP53*	*rs1042522*	x	x	x	x	x	x	x	x	x	x	x	x	x		x	x	x				x	x	x	x	x	x	x	x	x	x	x	x	x	x	x	x	x	x	x	x	x	x	x	x	0.33
	*rs28934578*						x																																							0.000004
	*rs587782006*																						x																							0.000004
*PIK3CA*	*rs780807237*																													x																0.000005
	*rs2230461*				x	x	x												x	x	x		x																							0.06
*KDR*	*rs1870377*				x	x	x						x	x					x	x	x				x	x	x											x	x	x	x	x	x	x	x	0.22
	*rs397772062*														x																															0.22
*JAK3*	*rs3213409*				x	x	x																																					x	x	0.008
*SMARC1/DERL3*	*rs5030613*						x																					x	x	x	x	x	x													0.144
*KIT*	*rs3822214*							x	x	x			x	x								x	x	x																						0.09
*NPM1*	*rs34323200*														x																															0.000004
*APC*	*COSM5990839*																				x																									-
	*Frameshift Deletion*					x																																								-
	*rs121913329*																						x																							-
*FGF3*	*rs17881656*																					x	x	x																						-
	*rs149119664*																						x																							-
*MLH1*	*rs876659657*																						x																							-
*FLT3*	*rs752729752*																						x																							0.0001
	*rs2491231*	x	x	x	x	x	x				x	x	x	x	x			x	x	x	x	x	x	x	x	x	x	x	x	x	x	x	x					x	x	x	x	x	x	x	x	0.4
	*rs199906407*			x																																										-
	*rs75580865*																							x																						0.05
*AKT1*	*rs990046031*																						x																							-
*STK11*	*rs587781474*																						x																							-
	*rs59912467*																																											x	x	0.006
*RB1*	*p.E137K*																																x													-
*RET*	*rs1800863*																							x																						0.17
*ABL1*	*rs34549764*																											x	x	x																0.002
*ATM*	*p.L2868F*																																x													-
	*rs1800056*																											x	x	x																0.01
*HRAS/LRRC56*	*rs28933406*																												x	x																-
*CTNNB1*	*rs121913403*																															x	x													-
*MTTP*	*rs748883732*																			x																							x			
*EGFR*	*rs766533982*																																	x	x											0.000008

dbSNP = single nucleotide polymorphism database; H = healthy tissue; T = tumor tissue; M = metastasis tissue; MAF = Minor Allele Frequencies; * = Stop Codon.

**Table 3 cancers-14-04212-t003:** *p*-values of the correlation test between all variables.

*p*-Value	Tropism	MAF	ClinVar	Gleason	PSA	MGS	Age	pT	N
Tropism		0.00013	0.07011	0.67739	0.86292	0.08165	0.59258	0.54789	0.00199
MAF	0.00013		0.48214	0.77437	0.87733	0.28901	0.11517	0.90036	0.06808
ClinVar	0.07011	0.48214		0.8074	0.95044	0.24062	0.16091	0.2357	0.0649
Gleason	0.67739	0.77437	0.8074		0.78992	0.80144	0.25762	0.65732	0.82564
PSA	0.86292	0.87733	0.95044	0.78992		0.7361	0.79574	0.25965	0.80498
MGS	0.08165	0.28901	0.24062	0.80144	0.7361		0.94487	0.85406	0.26163
Age	0.59258	0.11517	0.16091	0.25762	0.79574	0.94487		0.64361	0.90357
pT	0.54789	0.90036	0.2357	0.65732	0.25965	0.85406	0.64361		0.86292
N	0.00199	0.06808	0.0649	0.82564	0.80498	0.26163	0.90357	0.86292	

## Data Availability

The raw data supporting the conclusions of this article will be made available by the authors, without undue reservation.

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
