# Peer review of "Genomic Landscape Alterations in Primary Tumor and Matched Lymph Node Metastasis in Hormone-Naïve Prostate Cancer Patients"

_cancers, 2022, doi:10.3390/cancers14174212_

Round 1

Reviewer 1 Report

1. The comparison of somatic mutations from the primary tumour and metastasis is most relevant since targeted agents such as PARP Inhibitors (depending of region of approval) for HRD mutated tumours are available. The PROfound study showed already that NGS from the paraffin embedded primary tumour is sufficient and no need for biopsies of the metastases is generally needed. The authors should include these data also in the background of the manuscript. 

2. The authors present a relevant, though very small patient cohort with NGS performed from the primary and the lymphe node metastasis. The authors should provide a table that highlights the timing of lymph node biopsy or operation (time from primary tumour to lymph node metastasis) and what treatment (systemic or local) was applied in these patients.

3. The panel of the NGS used is not used in daily clinical practice. It would have been more adequate to use the panel that includes the genes that were used in the PROfound phase III trial or is included in the comercially available NGS (Oncomine or Foundation One) since relevant HRD alterations such as CDK12 for example was not inlcuded in the author's panel. This information is missing and can not be provided, thus would have improved significantly the quality of the data reported. 

4. The correlation of specific alterations and mutations with overall survival is only hypothesis generating and needs a multivariat analysis or Cox-Regression Model, since multiple biological factors will influence outcome as well as systemic or local treatment applied. If data cannot be provided the authors should discuss these limitations vigorously. The statistical methods do not indicate any details on how survial curves were assessed and calculated. This must be verified. 

Author Response

We would like to thank you for your comments and time dedicating in revising our paper.

Please find below our replies:

  1. We agree with your observation, however the scope of our study was a bit different from the PROfound study. In particular, we aimed to verify the synchrony of alterations in primary and metastatic tissue, for this reason we had to collect both tissues.
  2. We agree with your observation and we apologize for the lack of better specification. Patients received simultaneously surgery + lymph node dissection (at the time of surgery). We updated the methods. For this reason, we did not insert a table reporting follow-up for metastasis since patients with positive lymphnode received androgen deprivation therapy following surgery as per clinical practice.
  3. Many thanks for the observation, it’s true that the panel we applied is not used in clinical practice, but it’s very used in the research practice. This panel enables highly sensitive variant detection in a variety of cancer types, including prostate cancer. This panel was also used for many other studies (https://doi.org/10.18632/oncotarget.7343;https://doi.org/10.1007/s00428-017-2288-7; https://doi.org/10.1158/1538-7445.AM2019-3007). This paper is a part of a larger project, where we have tested another panel for the libraries creation (in this case from Qiagen) in combination with Illumina products for the dilution and for the sequencing (https://doi.org/10.1007/s00432-022-04262-0). Still, Hotspot Cancer Panel provided by Illumina are nowadays validated for the clinical practice (https://doi.org/10.5858/arpa.2013-0710-OA). It’s true that many private laboratories have the instrument for the diagnosis with new throughput technology, but rarely they have the scientific background to design custom panel. For this reason, this project is very important to study and in case modify or integrate the existents panels. Finally, the Oncomine Tumor-Specific Panels are designed for the Ion Torrent platform (https://www.ampliseq.com/otherContent/help-content/help_html/GUID-A6BF8309-005F-4307-B765-5D34AF478E94.html) and unfortunately we don’t own this technology, that’s why our choice fell on a kit that would allow us to use the MiSeq sequencer at our disposal. The data about quality are reported in the supplementary files.
  4. As concerning statistical analysis for survival, we used we used cBioPortal (https://www.cbioportal.org/) to verify the impact of found alterations on survival. cBioPortal provides a visual tool for research and analysis of cancer gene data and helps cancer tissue and cytology research gain molecular data understanding of their genetics, epigenetics, gene expression and proteomics. Through cBioPortal online tool, we analyzed gene alterations and their correlations with survival. To this regard, we did not perform statistical analysis on our own and for this reason we did not report on “statistical analysis” section. We hope to have fulfilled your request. As concerning the use of multivariate model, it is not possible in our cohort. In particular, the low number of patients and low number of events make it impossible to insert independent variable (normally 1 for each 10 events). We updated anyway the limitations.

Reviewer 2 Report

    1-    Title: The title reflects the content well. Just remove the period at the end of the title. If authors see that adding ERBB4 to the title might better reflect the main findings, they can add it. Otherwise, they can leave the title as it is.

2-      English: The manuscript could benefit from editing for grammar, missing words, and subject-verb agreement, etc. It is recommended that authors delete irrelevant "general" phrases and sentences, repeated and unneeded words. They should use short sentences. Also, some Introductory sentences are irrelevant or are not needed. There are also typos in the manuscript. For example, in the introduction, “among risk factors” should be “among the risk factors.” Also, use of punctuations needs to be revised: “i.e. diet, tobacco smoking, obesity” should be written as “i.e., diet, tobacco smoking, and obesity.” Another example is “Mutated) o HOXB13 (Homebox B13)” that should be written as “Mutated), or HOXB13 (Homebox B13).”

3-      Simple summary: “We found mostly concordance between both primary PCa samples and matched lymph node metastasis in terms of genomic alterations, underlining those alterations in primary tumor are extremely important for cancer prognosis prediction.” I suggest specifying important and key alterations rather than being general and just say alterations.

4-      Abbreviations: All abbreviations should be revised and defined at their first use.

5-      Abstract: “We retrospectively analyzed data from 470 patients who underwent surgery for PCa between 2010 and 2020 at the Department of Urology, University of Catania. The final cohort consisted of 17 different patients.” Readers might wonder how 470 patients were reviewed and only 17 were included. Probably, mentioning briefly the inclusion criteria here will help reviewers have a synopsis of why only 17 patients were included.

6-      Abstract: in the conclusion, authors can specify ERBB4 as the mostly mutated gene that was found to be concordant between primary samples and lymph nodes.

7-      Introduction: “PCa is still the second leading cause of cancer death worldwide.” Please add reference (Siegel et al., 2022).

8-      Introduction: with respect to lymph node metastasis in PCa, studies show that it does not affect prognosis. In fact, lymph node dissection in radical prostatectomies in a debatable issue recently. Authors can highlight on this topic briefly and suggest why it is important to do the study they did as regards understanding the genomic landscape of primary tumor versus lymph nodes and how this will help researchers better decipher the mechanisms responsible for lymph node metastasis.

9-      Methods: authors should include the inclusion criteria they used which decreases their cohort from 470 to 17 patients.

10-  Methods: “For each patient, an expert pathologist marked on hematoxylin and eosin-stained sections primary PCa tissue, a marked areas of distant adjacent normal tissue and lymph node metastasis tissue (in positive patients).” This statement needs rephrasing and correction of English language. Besides, please specify whether the pathologist is a specialist GU pathologist or general pathologist. Readers might wonder what area did the pathologist mark? Is it the area with highest grade group?

11-  Methods: please add details regarding the dilution factor of all antibodies.

12-  Methods: authors did not upload Supplementary Table 1 and Supplementary file 2. Please upload them. Authors need to be consistent with naming the supplementary material. What is Supplementary file 1 if there is file 2? Is it the Table?

13-  Methods: with respect to the statistical analyses, I am not an expert so I could not give my opinion in that.

14-  Results: Title of Table 1 needs revision: “Baseline characteristics of patients.” Remove “lists the.”

15-  Results: Why is the pathologic N stage not included in Table 1? Biopsy ISUP score should read ISUP Grade Group and not score. This is confusing as Gleason score is different from Grade Groups.

16-  Results: Authors need to add a small paragraph before Table 1 summarizing the findings and citing Table 1.

17-  Results: Formatting of Table 1 needs to be revised. Titles of variables can be in bold, and their categories below them not bolded. Authors can refer to this paper to have an idea how to better represent their results: Tables 1, 2, and 3 in https://linkinghub.elsevier.com/retrieve/pii/S1092-9134(21)00024-1.

18-  Results: In Table 2, why did authors use S=mucosa? It has been mentioned earlier that authors compared the tumor tissue to adjacent non-tumor tissue (not mucosa).

19-  Results: In scientific writing, in general, symbols for genes are italicized whereas symbols for proteins are not italicized. The formatting of symbols for RNA and complementary DNA (cDNA) usually follows the same conventions as those for gene symbols. Gene names that are written out in full are not italicized (e.g., insulin-like growth factor 1). Genotype designations should be italicized, whereas phenotype designations should not be italicized. For example, in Table 2, all genes need to be Italicized.

20-  Results: What is MAF in Table 2. Please add the definition of all abbreviations below Table 2. In fact, the formatting of this Table also needs revision. The numbers 1 to 17 refer to patients but this should be made clear where a title for this row can be added to the left saying “Patients.”

21-  Results: “The patient where were found most non-synonymous dbSNPs is Patient 8.” This statement is unclear. Authors cannot put “where” then “were.” Please revise.

22-  Results: what is “fastq” on line 194?

23-  Results: again, I believe saying mucosa when describing the results is confusing. Did authors take adjacent normal prostate tissue or mucosa?

24-  Results: authors have been using MAF without defining it. Do they mean minor allele frequency?

25-  Results: Format of Table 3 needs revision. What is VS? do they mean versus? If so, write the whole word and avoid using abbreviations that will confuse readers that might not be aware of what authors mean.

26-  Results: in Table 3, what is the p value presented? How is it in negative? Also, why are there blue and red colors? Do they mean anything?

27-  Results: Supplementary Table 4 is not uploaded. Also, is there Supplementary Table 3? Authors jumped from Table 2 to 4 with no Supplementary Table 3.

28-  Results: Specify in the results section what were the databases that Supplementary Figures 1-5 patients were taken from.

29-  Results: I am wondering about the race of the patients included. This might need to be specified as genetics differ with ethnic groups and findings might be only applicable to this population of patients.

30-  Figures: All the figure legends can be revised as to be more informative of the images presented. Also, statistical tests used and meaning of asterix need to be added. Abbreviations used withing Tables and Figures should be defined as well in the legends at the end.

31-  Discussion: Authors should focus more on the main findings and avoid repeating results presentation in the discussion. Authors could also correlate their findings with what has been published in literature. Clinical relevance should be added.

Author Response

We would like to thank you for your comments and time dedicating in revising our paper.

Please find below our replies:

    1. We thank you for the comment. We have removed the period and left the title as it is.
    2. We thank you for the corrections. We have revised the paper in its English form. After the potential acceptance we will do extensive english revision using MDPI service.
    3. We updated the simple summary.
    4. We revised all abbreviations.
    5. We revised the abstract and methods by adding: Inclusion criteria were: patients with lymph node metastasis and patients with PCa with extra capsular extension (pT3) and negative lymph node metastasis.
    6. We corrected as per your suggestion.
    7. We added.
    8. As concerning this regard, we are not completely in agreement. For example, in a recent paper (DOI: 10.1016/j.juro.2017.12.048) authors demonstrated that the risk of cancer specific mortality significantly increased in men diagnosed with pelvic lymph node metastases (HR 4.5, 95% CI 4.2-4.9, p <0.01) and distant metastases (HR 21.9, 95% CI 21.2-22.7, p <0.01) compared to men with nonmetastatic disease. We updated the manuscript for a better specification of our aims.
    9. We updated the inclusion criteria.
    10. Thank you for this comment. We updated the sentence.
    11. Thank you for your comment. We used manufacturer instructions for all dilution and to this regard we reported code for each Ab and added dilution. We updated the text.
    12. We apologize for the misunderstanding. The supplementary file contained all suppl. tables including the number 1 until number 4.
    13. We would like to thank you.
    14. We made the correction.
    15. We made the correction.
    16. We made the correction.
    17. We made the correction.
    18. We modified S (sane, mucosa) with H (healthy tissue), that is more general.
    19. We made the correction
    20. We modified table 2.
    21. We modified the text.
    22. We corrected the sentence.
    23. We modified S (sane, mucosa) with H (healthy tissue), that is more general.
    24. Yes, we updated the abbreviation in the all text.
    25. Yes, we corrected the table.
    26. The P is the pearson correlation. We updated the table 2.
    27. We updated the manuscript. We reported all supplementary tables.
    28. Thank you for this comment. We have updated the manuscript.
    29. All patients were European-Caucasian. We have modified the manuscript.
    30. We have added a legend in all tables.
    31. We have removed the sentence in the discussion that repeated the results.

Reviewer 3 Report

In this manuscript, the authors report their findings on association of mutations with lymph node metastasis of prostate cancer in clinical cases of prostate cancer. The study is interesting but lacks data in Androgen Receptor (AR), a major player in prostate cancer. Besides, the authors are encouraged to provide IHC images as main figures. Some of the relevant images from supplementary data may be moved to main figures.

AR is known to mutate in prostate cancer patients. In fact, AR and its splice variants play significant roles in driving prostate cancer progression. The authors are encouraged to focus on this aspect while consulting and citing most recent works like the one by Thomas et al, 2022.

Thomas E, Thankan RS, Purushottamachar P, Huang W, Kane MA, Zhang Y, Ambulos N, Weber DJ, Njar VCO. Transcriptome profiling reveals that VNPP433-3β, the lead next-generation galeterone analog inhibits prostate cancer stem cells by downregulating epithelial-mesenchymal transition and stem cell markers. Mol Carcinog. 2022 Jul;61(7):643-654. doi: 10.1002/mc.23406. Epub 2022 May 5. PMID: 35512605; PMCID: PMC9322274.

Author Response

We would like to thank you for your comments and time dedicating in revising our paper.

Please find below our replies:

  1. We agree with your observation. However, in our results we did not find significant data about AR. We added as potential limitation. We have added some relevant figure and we have also added as supplementary the staining figures of Ab.
  2. We would like to thank you about this comment. We have updated the introduction by adding your comment.

Reviewer 4 Report

The present manuscript identifies genomic alterations associated with lymph node metastasis in a cohort of hormone-naïve prostate cancer patients. Furthermore, authors describe concordance between both primary PCa samples and matched lymph node metastasis. In my opinion, the study looks interesting, brings novelty in the potential use of sequencing high-throughput technologies and does provides some interesting data related to cancer prognosis prediction which could be also applied for monitoring disease and drug response after curative treatment. There are however, few clarifications and modification required before further consideration.

Questions and concerns to be addressed properly:

General:

- Acronyms should be explained when they first appear. Author should revise this aspect on the whole manuscript. 

1. Introduction:

-Author need to restructure the different paragraphs, as it seems that each sentence completes a paragraph. Sentences related to the same idea need to be grouped in the same paragraph to improve the comprehension of this section.

2. Methodology:

- The first section of the material and methods is related first to characteristics of the used cohort and then there is a long explanation about different experiments carried out with the studied PCa tissues, such as, hematoxylin and eosin stain, immunohistochemical analysis, the scoring. Maybe all this section needs to be named as the rest of the sections in the material and methods (2.1 or 2.2 or 2.3…), with its title.

- In line 70-71, I do not understand the difference between patients with lymph node metastasis and patients without negative lymph node metastasis.

- All the times that antibodies are used, the used dilution should be added together with the commercial house and reference.

- In section 2.2 NGS Sequencing, authors refer “all samples were diluted to a 5ng/ul concentration and then 20 ng of each sample have been used for sequencing”. Authors should clarify if there is DNA (as I could understand with the previous section) or what type of sample they are using.

- In section 2.3. Bioinformatic analysis authors describe the use of FASTQ format files for mutation detection. They describe very slightly, which tools and applications have used for the analysis but there is a lack of detailed information if readers want to reproduce the experiment. Authors should add specific links or references of used tools and applications, the used parameters for each tool…. The same happens in section 2.4 when describing the methodology used with cBioPortal analysis, it needs to be extended adding used parameters…

-In section 2.4, authors describe how they calculated the multiparameter genetic score. Has been this score previously validated in other publications? How do authors verify that is a good score to use in clinics?

-In section 2.5 Statistical analysis, authors describe how the doy the correlation among the scores and different clinical and non clinical data. Maybe it is appropriate to describe those clinical and non clinical data or the parameters that were measured to the patients, samples etc, in the first section of the materials and methods where the cohort is described.

3. Results:

-As a title of each section of Results, I usually appreciate that each title summarizes the general result or the general conclusion of what is being explained, rather than the used technique name. This helps the reader to clarify concepts and have the most important idea of the section in mind. Therefore, authors should try to change the titles to catch reader’s attention.

- The Results should not start with a table that is not explained in nowhere. The section should start introducing the first results that in this case are de baseline characteristics of patients. As I have mentioned before, this measured parameters should be explained/listed also at the beginning of materials and methods section, when the study cohort is described.

- All the figure and tables should have the explanation of the acronyms used in each figure or table.

- In the non published material files I only found the supplementary figures but not the supplementary tables that I would like to review.

- In section 3.2 multiparameter genetic score, if I have not understand it wrong, there is a description of the supplementary table 3, but there is not a description of the main results of Figure 1, which is added next to the text. Authors should also describe the figure 1 and the most important results.

-In section 3.4, I do not understand why authors put attention in studying more in deep only genes ERBB4, HRAS, KIT, ABL1 and CTNNB1. Maybe an explanation about why authors select these genes should be added.

-Again in section 3.5, I do not understand why authors put attention in studying more in deep the statistical association between IGF1-R and ERRB4 (and also AR, IR-alfa, IR-beta, and PSMA). Furthermore, there is no table or figure showing those results.

-To sum up for Results, I see the lack of a good structure when describing the observed results. There is a need to tell a story, which links one results with the next, telling the different goals proposed and the obtained results for each section.

4. Discussion:

- In general, discussion reads more like a review of what has been investigated previously. It seems that the real discussion of the obtained results does not start until line 306. In my opinion authors need to rewrite the discussion, presenting the main results and comparing them with similar works and not just a description about what it has been done previously followed by a repetition of the obtained results.

- From what I understand, I think that in line 283, “Schmidt et al we analyzed laser micro-dissected primary …..” that “we” needs to be remove.

-In my opinion more about this scores and results related to clinical use could be discussed, focused on the utility of the obtained results, to give more importance to the manuscript. Furthermore, this is mentioned at the final sentence of the conclusions, so I think it could be extended in the discussion.

-As authors mentioned in the introduction, PCa risk factors are associated with lifestyle. The obtained results in those mutations could be linked with patient’s lifestyle?.

-Limitations are repeated twice, so authors should delete the text from line 337 to line 339.

Author Response

We would like to thank you for your comments and time dedicating in revising our paper.

Please find below our replies:

  1. We have corrected the text regarding all abbreviations
  2. We modified the introduction according to your suggestion
  3. We modified the text based on your suggestion.
  4. We apologize for the misunderstanding.

We corrected as following: “Inclusion criteria were: patients with lymph node metastasis and patients with PCa with extra capsular extension (pT3) and negative lymph node metastasis. The final cohort consisted of 17 different patients (11 PCa with lymph node metastasis and 6 PCa without lymph node metastasis)”

  1. We have updated by adding dilution.
  2. We apologize for the misunderstanding. Yes, it was intended the DNA. We updated the relative part.
  3. Thank you for your comment. We didn’t insert the full procedure just because this is whole reported in the Illumina Basespace Sequence Hub and in the application “DNA amplicon”, and so it could be redundant. Furthermore, the procedure is long and insert it in the text could compromise the linearity of speech. Despite this, we specified in the text that we followed the instructions reported in the site.
  4. We updated the manuscript regarding the use of the cbBioPortal.
  5. We agree with your observation. Since there are no other similar score, in order to improve the manuscript we constructed this tool. We agree with your observation and we updated the limitation in the manuscript since we lack of validation.
  6. We updated this part.
  7. We agree with your observation and we changed all sub-titles.
  8. We followed your suggestion and we made the correction.
  9. We added caption in all figures
  10. We gave more attention in uploading all suppl. files.
  11. We made a correction in the text according to your suggestion
  12. We apologize for the lack of better explanation. As concerning section 3.4 we studied those alterations that were expressed both in primary tumor and in matched lymph node metastasis. We made a correction in text.
  13. We agree with this consideration. We added this part in the introduction: “It is also important to underline that in the last years, many studies have highlighted the hypothesis that diabetes and hyperglycemia could be relevant for PCa development and progression and in some cases with worse recurrence over time 8. In a previous study from Broggi et al it has been reported that androgen receptor (AR) expression was associated with increased risk of pathological aggressiveness (odds ratio [OR]: 2.2; P <0.05) and insulin-receptor-α (IR- α) expression (OR: 5.7; P <0.05) and that insuling growth factor-1 (IGF-1) expression was predictive of pathological aggressiveness (OR: 16.5; P =0.017) in patients with PCa and diabetes. Up to date, the genetic understanding of patients with more advanced disease and local metastasis and to evaluate potential driver of PCa progression”.
  14. We modified the results section
  15. We modified the discussion
  16. We corrected the typo.
  17. We modified the discussion according to your consideration.
  18. We agree with your observation and we removed that sentence.
  19. We removed the duplicate.

Round 2

Reviewer 2 Report

Thank you.

Author Response

I would like to thank you for your precious comments. 

Reviewer 3 Report

The authors have addressed the points mentioned in the previous review and the revised manuscript is significantly improved. However, there are few instances of typo such as 'Line 394, Androgen Receptor' and elsewhere.

The manuscript may considered for publication after spelling check.

Author Response

We would like to thank you. We have revised again the paper. 

Reviewer 4 Report

I appreciate the efforts of the authors to be able to satisfy the changes proposed by the reviewers in such a short time. However, a few more changes and questions are proposed before further considerations:

Questions and concerns to be addressed properly:

1. Methodology:

- All the times that antibodies are used, together with the used dilution, also the commercial house and reference should be added.

 In section 2.3. Bioinformatic analysis authors describe the use of FASTQ format files for mutation detection. They describe very slightly, which tools and applications have used for the analysis but there is a lack of detailed information. In my opinion authors should describe the procedure briefly specifying the most important parameters and also add the link of the used Ilummina protocol, to be accessible to all the readers. The same for section 2.4 when describing the methodology used with cBioPortal analysis.

2. Results:

-As a title of each section of Results, I usually appreciate that each title summarizes the general result or the general conclusion of what is being explained, rather than the used technique name. This helps the reader to clarify concepts and have the most important idea of the section in mind. Authors have response that they have change those titles, but the true is that they are similar or the same as the first version.

- In my opinion, more extended description of the first table needs to be developed, putting attention to the interesting parameters that authors have measure to the patients (not only two).

- As I said in the previous revision, in section 3.2 multiparameter genetic score, there is not a description of the main results of Figure 1, which is added next to the text. Authors should also describe the figure 1 and the most important results observed there. Also the table 3 has not table title.

-In section 3.4, authors have explained in this second version why they have analyzed ERBB4, HRAS, KIT, ABL1 and CTNNB1, but I think the sentence needs to be linked in a better way with the next sentence in the text.

-Authors have added an explanation in the introduction about why they study the statistical association between IGF1-R and ERRB4 (and also AR, IR-alfa, IR-beta, and PSMA) in section 3.5. In my opinion a brief recordatory of that explanation should be added also in section 3.5.

3. Discussion:

-In my opinion more about this scores and results related to clinical use could be discussed, focused on the utility of the obtained results, to give more importance to the manuscript.

-As authors mentioned in the introduction, PCa risk factors are associated with lifestyle. The obtained results in those mutations could be linked with patient’s lifestyle? I think authors did not answer to this question in the second version.

Author Response

We would like again to thank you again for your time dedicated in improving our manuscript.

Please find enclosed our comments:

  1. We have added a reference in after each Ab.
  2. We have improved the related sections as per your suggestion.
  3. We better understand your suggestion. We have changed the relative subtitles.
  4. We added a sentence more. A part of this there is no so much to tell about the table that only describes all clinical and pathological characteristics of the cohort.
  5. We made the corrections as you suggested.
  6. We agree with your observation and we made the correction.
  7. We made the correction you suggested.
  8. We agree with you and we updated the manuscript.
  9. We agree with your previous observation that the concept of lifestyle was not an outcome of the study. To this regard, we removed the sentence on lifestyle from the manuscript in order to avoid confusion. In the final manuscript there is no mention of lifestyle control.